# Choices of Specialties and Training Sites among Taiwanese Physicians Graduating from Polish Medical Schools

**DOI:** 10.3390/ijerph19063727

**Published:** 2022-03-21

**Authors:** Tzu-Ling Weng, Feng-Yuan Chu, Chiao-Lin Li, Tzeng-Ji Chen

**Affiliations:** 1Department of Family Medicine, Taipei Veterans General Hospital, Taipei 112, Taiwan; tlweng@vghtpe.gov.tw; 2School of Medicine, National Yang Ming Chiao Tung University, Taipei 112, Taiwan; fychu@vghtpe.gov.tw; 3Division of Clinical Toxicology and Occupational Medicine, Department of Medicine, Taipei Veterans General Hospital, Taipei 112, Taiwan; 4Department of Family Medicine, Taipei City Hospital Yangming Branch, Taipei 111, Taiwan; dbc57@tpech.gov.tw

**Keywords:** career choices, international medical graduates, Polish medical school, residency, specialty, Taiwan, urbanization

## Abstract

Taiwanese students who graduated from Polish medical schools (P-IMGs) accounted for the second-largest group of international medical graduates in Taiwan. In 2009, domestic medical students in Taiwan staged mass demonstrations against P-IMG’s exemption from the qualifying test before the licensing exam. Although medical circles in Taiwan might still hold prejudices against P-IMGs, little is known about their career development. This study will analyze P-IMGs’ choices of specialties and training sites from 2000 to 2020 using data from the membership section of the Taiwan Medical Journal, the monthly official publication of the Taiwan Medical Association. Of 372 P-IMGs, 34.2% chose internal medicine and 17.1% surgery. Although academic medical centers offered 76% of all available trainee positions in a year, only 49.3% of P-IMGs received training there. By contrast, 20.9% of P-IMGs were trained at nonmetropolitan hospitals that altogether accounted for only 5.8% of trainee positions. In conclusion, P-IMGs had their residency training at less favorable specialties and sites. Their long-term career development deserves further study.

## 1. Introduction

The term “international medical graduates” (IMGs) is generally understood to denote a physician who is awarded their medical degree in a country other than the one where they intend to practice medicine [1,2,3]. Over recent decades, IMGs have played a significant and expanding role in medical practice in many developed countries throughout the world, constituting approximately 31% of physicians in the United Kingdom, 25% in the United States, 17% in Australia, and 23% in Canada [1,4]. The wave of globalization, an aging population with increasing demands for medical care, and the pursuit of a better quality of life could all be reasons for this phenomenon [5,6,7]. Additionally, IMGs can be divided broadly into two categories. The first comprises those physicians who graduate from native or foreign medical schools and who practice in foreign countries, like most of those IMGs practicing in Western Europe, Canada, and the United States [8,9]. The other, which we call “native IMGs,” are those physicians who graduate from medical schools abroad and return to practice in their home countries. Most IMGs in Taiwan fall into the second category [10].

### 1.1. Taiwan Medical Education System

Taiwan’s medical education system has been revamped in recent years: 6 years of medical school after senior high school, including 2 years of clinical clerkship. Passing the national examination for physicians is followed by 2 years of postgraduation training and then residency training in various specialties [11,12].

Taiwan has 12 medical schools, four public, and eight private, with the capacity for 1450 medical students per year, which the government controls to regulate the final supply of physicians. However, compared with the number of high school graduates each year, only the top 1% are likely to manage to squeeze through the narrow gate of the medical hall. This fact and the belief that doctors are well paid and have high social status still prevail in our society [13]. Many students study medicine in other countries but eventually return to their home country to practice after graduation.

### 1.2. Taiwan’s IMGs

According to statistics, IMGs have constituted approximately 3.7% (approximately 1800 doctors) of the workforce in Taiwan in recent years [10]. In 2018, the country where most IMGs studied medicine was the Philippines (31.7% of total IMGs), followed by Poland (24.2%) and Myanmar (21.0%). It was primarily native IMGs, Taiwanese, who got medical degrees from the Philippines and Poland. There is a long history of Taiwanese students studying medicine in the Philippines, but it is only in the past 20 years that they have been going to Poland. In the following, we define Taiwanese students who graduated from Polish medical schools as “P-IMGs”. In 2008–2009, the conflict over Polish medical degrees and the recognition of qualifications erupted.

### 1.3. The Debate over P-IMGs

The Taiwanese government has established a system whereby IMGs must pass a degree-verification exam (DVE) to be equal to Taiwanese medical graduates (TMGs) and are required to take the national licensing examination (NLE). Since 2002, considering that several countries in the world had advanced medical quality and education, IMGs who had medical degrees from nine countries or areas (including the United States, Japan, Europe, Canada, South Africa, Australia, New Zealand, Singapore, and Hong Kong) were excluded from the previous policy; that is, they had been allowed to bypass the DVE and entered the LE directly, similar to domestically trained TMGs.

Since 2004, several Eastern European countries have successively joined the European Union (EU). Although “Europe” in the nine privileged regions initially referred to the developed countries of Western Europe, Eastern European countries were also included after joining the EU. Poland is one of the few countries that offer English language programs for international students to study medicine [14,15]. There are 15 medical schools that offer 6-year English medical programs for international high-school graduates [16,17,18]. Compared with those in the US and Europe, their medical schools targeting international students have low entrance requirements and much lower tuition fees, coupled with a lower cost of living [14,19,20]. Attracting more than 7200 students from 170 countries in 2018 [21]. Together with the advantage of not needing to take the DVE, increasing numbers of students are choosing to study medicine in Poland, to avoid the narrow path in Taiwan, and eventually return home to practice. Even students who studied in other countries where they were required to pass the DVE, such as the Philippines, transferred to Poland.

In 2008–2009, approximately 50 P-IMGs returned to Taiwan to practice. However, some cases were reported wherein some of them were being questioned by co-workers and fellow doctors about their lack of clinical experience after joining the hospital staff [22]. Taiwanese medical students (TMSs) initiated a series of controversies over what they saw as a “loophole” that had emerged—they believed there was a lack of sufficient internship experience in Poland [23]. They requested the government to amend the law, hoping to let P-IMGs pass the DVE and fulfill a year of internship in a designated hospital before taking the NLE. The debate was quite fierce online; moreover, a street protest by 2000 TMSs broke out on 31 May 2009 [17,24]. In response, the Taiwanese government initiated a series of meetings and amended policies [25]. The latest version of the related law is that all IMGs, including those from the nine initial EU areas, must pass the DVE and have a full year of internship in our country before taking the same NLE as the other TMGs. The only exception is that if an IMG has a medical degree from one of the nine EU areas, has the qualifications to practice there, and has local practice experience over 3 years, they can bypass the internship and DVE and take the NLE directly [26,27].

Most of the previous studies on IMGs were about non-native IMGs and concerned their culture and language adaptation, performance on the qualification tests, the demographics of the IMGs, etc. Relatively little research has focused on native IMGs. We found few mention native IMGs in Canada, similarly facing more difficulties than Canadian medical school graduates while applying for jobs [28,29].

Only one existing study discusses P-IMGs in Taiwan. In 2014, Dr. Ming-Jung Ho discussed fairness and equality between TMGs and P-IMGs and the different policies between countries [17]. The numbers of P-IMGs have increased year on year since 2007 and reached 420 practicing physicians in total in 2018, yet we still know very little about their general situation over recent years. Even though more than a decade has passed since the controversy, without really understanding the issues, the medical community and the general public in Taiwan still raise questions about P-IMGs’ medical ability from time to time.

Hence, to fill this gap in our knowledge, the purpose of this study was to provide some indicators of the P-IMGs in terms of their roles in Taiwan’s medical society, by collecting and analyzing P-IMGs’ medical practice dynamics data from 2000 to 2020. The specific aim is to evaluate P-IMGs’ choices of specialties and working places and to see if there is any differences from those of domestic TMGs. Through our research, we hoped to build a new perspective on P-IMGs, who could be seen as fresh activists, while reducing the misunderstanding and prejudice to a certain extent. We expect more cooperation and communication to emerge between P-IMGs and TMGs.

## 2. Materials and Methods

### 2.1. Background

Because of the growing number of native IMGs year on year, the dispute between P-IMGs and TMGs in 2008–2009, and the misunderstanding over P-IMGs ever since, we aimed to analyze the characteristics and roles of P-IMGs in Taiwan’s medical society over the last 20 years.

### 2.2. Data Collection

This study collected information from the membership section of the Taiwan Medical Journal. According to history and previous research, P-IMGs first emerged in Taiwan around 2003; therefore, we decided to collect data from January 2000 to August 2020. The Taiwan Medical Journal is the journal published by the Taiwan Medical Association since 1958, with 12 issues published every year. Nearly all the content has been freely accessible online since the publication of Volume 43 in 2000. However, the “Members Activity” section was not published online until October 2019, in Volume 62. We collected the “Member Activities” section of those journals published since October 2019 from the Taiwan Medical Association’s website (https://www.tma.tw/magazine/index.asp (accessed on 14 March 2022)), and those published between January 2000 and October 2019 were extracted from the paper journals stored in the library of Taipei Veterans General Hospital.

The information in the section of “Member Activities” includes the doctor’s name, graduation school, the membership dynamics with each county’s physicians’ association (including different dynamics such as admission, withdrawal, change, suspension, reinstatement, cancellation, or death), the medical institutions they worked at, and the workplace they registered. When a member’s status changes, this information will be displayed only with the subject’s consent. How this journal shows the name of the graduate school should be expressly stated: if it is a foreign medical school, it will be represented by the name of the country. For example, only “Polish Medical School” or “American Medical School” is displayed instead of a specific school name, such as “Poznan Medical School” or “Harvard Medical School.” We manually searched and inputted all the data with “Polish Medical School” as the practitioner’s graduate school into a Microsoft Excel worksheet. The search was repeated by Dr. Weng and Dr. Lee in case any information was missed.

We also used the statistical data accessed similarly from the website of the Taiwan Medical Association (https://www.tma.tw/stats/index_AllPDF.asp (accessed on 1 February 2022)). They have provided several statistical works over several details of Taiwan’s medical system. We collected data from the charts named Statistics of Educational Background of Practicing Physicians.

To show the overall situation in Taiwan, we also referred to the training capacity of each hospital and specialty department as set out by the Department of Medical Affairs of the Ministry of Health and Welfare in Taiwan in 2020, that is, the number of trainees that can be accommodated (https://dep.mohw.gov.tw/DOMA/fp-2713-46862-106.html (accessed on 1 February 2022)). The training capacity of each specialty in each hospital depends on the size of the hospital, the number of physicians available to work as teachers, the training situation in the past, etc. When interpreting the information comprehensively, in terms of being regionally oriented or specialties oriented, it can to some extent reflect the popularity of a specialty, the demand for it, and the willingness of medical graduates to choose this specialty.

### 2.3. Research Design

After inputting the fragmented data from the journals, we then reshaped them into dynamic trends of each individual also using Microsoft Excel. Not every dynamic of P-IMG’s practice is intact, as each piece of information is presented or not, depending on their own will.

For comparison, because it is challenging to obtain only the information of TMGs and exclude each IMG, in this study, we decided to use the training capabilities of each specialty to provide some clues for the overall choices of medical graduates. Although these statistics are based on the general medical condition in Taiwan, including all IMGs, and because IMGs only account for 3.7% of all physicians and TMGs account for more than 95% of total physicians, we used these data to some extent as representing the situation of TMGs to compare with that of P-IMGs.

We also analyzed the regional distribution of P-IMGs’ specialist training according to the urbanization level of the area and the level of hospitals they trained in. The degree of urbanization of cities and towns in Taiwan is classified into seven levels based on their demographic characteristics, industrialization, and medical resources [30]. Level 1 represents the highest population density, level of education, and density of medical resources, whereas Level 7 represents the lowest levels in the overall evaluation. We defined areas with Levels 1 and 2 as urban areas, Levels 3 and 4 as suburban, and Levels 5–7 as rural areas [31].

The hospital level, classified into medical centers, regional hospitals, and local hospitals, is decided every 4 years by the Joint Commission of Taiwan through hospital accreditation. We checked each hospital level online through the 2017–2020 qualified list (http://service.jct.org.tw/TJCHA_CERT/ha.aspx (accessed on 1 February 2022)).

### 2.4. Statistical Analysis

Only descriptive statistics were calculated. Data analysis was conducted with Microsoft Excel MAC version 16.58.

### 2.5. Ethical Approval

The data we used were all publicly accessible. With the Personal Data Protection Act and human research regulations in Taiwan, the Ethics Committee’s approval was not needed in this study.

## 3. Results

On the basis of the data collected from the journals, 372 P-IMGs were identified from January 2000 to August 2020, as shown in Figure 1. Of the total 372 P-IMGs, 135 (36.3%) were women, 232 (62.4%) were men, and five were unknown. The dramatic debate online and the protest were held mainly in 2009. In 2010, the lowest number of P-IMGs returned home; however, in 2013–2017, the largest number of P-IMGs returned, with approximately 44–71 P-IMGs per year, most markedly in 2013. The number of P-IMGs collected from the journal each year was found to be much fewer than the actual number recorded from the official website by the Taiwan Medical Association. Because the journal only displayed related information with the subject’s consent, it seems that a certain percentage of P-IMGs tend not to share their details with the medical society.

We noticed a significant difference when comparing the distribution of the first trained specialties of these 372 P-IMGs and the training capacity of each specialty in a year in Taiwan, as shown in Figure 2. In this study, we used the training capacity of each specialty to represent the overall trend and preference of medical students in choosing specialties and the true needs of each subject. We used data from 2020 as a representation. We found that the top five specialties that P-IMGs chose to be trained in were internal medicine (34.2%), surgery (17.1%), emergency medicine (10.3%), pediatrics (9.4%), and obstetrics/gynecology (9.0%), which accounted for 80% of P-IMGs in total; while in a year in Taiwan, the training positions of these five specialties account for only 56.5% of total residency training positions, 20.4%, 17.9%, 6.6%, 7.4%, and 4.2%, respectively. In contrast, specialties such as ophthalmology, pathology, rehabilitation, dermatology, and plastic surgery had a relatively small training capacity each year and had almost no P-IMG involvement.

Similarly, we also used the specialty training capacity in 2020 to in some way represent the general medical society while discussing the levels of the training hospitals and urbanization levels of the training sites. Table 1 shows the total P-IMGs collected from the journal, with 151 (49.3%) trained in medical centers, 150 (49%) trained in regional hospitals, and 5 (1.7%) in local hospitals. Approximately 79% of P-IMGs underwent their first specialty training in urban areas and 21% in non-urban areas, while the general medical graduates had a slightly different distribution, as shown in Table 2. We found that 94.2% of training positions were in urban areas and 5.8% were in non-urban areas and that 76% of the positions available were in medical centers, 23.7% in regional hospitals, and 0.4% in local hospitals.

## 4. Discussion

To sum up, this study analyzes the choices of P-IMGs in specialist training and their similarities to and differences from those of general medical graduates. Our findings found that P-IMGs chose specialties differently from TMGs, and a higher proportion of P-IMGs chose to receive residency training in regional hospitals and urban regions than TMGs.

The results indicate that the career choices of P-IMGs are slightly different overall from those of domestically trained medical graduates. It can be reasoned that they have different medical education backgrounds or face unequal opportunities. Moreover, a percentage of them tend to conceal their information from the public, possibly due to social distrust.

Our study discovered a higher ratio of P-IMGs practicing in internal medicine, surgery, emergency medicine, pediatrics, and obstetrics/gynecology. In recent years, these specialties have been considered less attractive in Taiwan [32,33,34]. Since the National Health Insurance system reforms carried out from 1994 [35], and because of the low insurance costs and high-quality medical care that everyone is eager to maintain, the working hours of physicians have been overextended [36,37], and the salary has been disproportionate [38,39]. Additionally, distrust in the doctor–patient relationship has led to more and more medical lawsuits [40,41,42,43]. With this background, along with the aforementioned specialties having higher risk, more brutal working content, and longer working hours, they have been less attractive to medical graduates making career choices in recent decades [44].

However, P-IMGs seemed to be more likely to join these specialties and fill the critical gaps, as with those in the US [45]. We found papers stating that IMGs in the US tend to practice in primary care specialties [46,47,48], as did foreign medical graduates in Australia [4]. Another paper found that IMGs practiced more in four specialties: internal medicine, neurology, psychiatry, and pediatrics [49]. Moreover, IMGs in Canada are more willing to choose family medicine [50]. Our findings do not contradict those results.

A partial explanation for our results may be due to those highly competitive specialties in Taiwan, such as dermatology, plastic surgery, ophthalmology, and rehabilitation, being more competitive [51]. With many other outstanding candidates, P-IMGs’ academic background may not be to their advantage in the current social atmosphere. Our results show nearly no P-IMGs have chosen these specialties. This result may also confirm our conjectures. The other possible explanation is that the healthcare situation is quite different between Poland and Taiwan. Poland’s health system is excessively concentrated on in-hospital care [52] and those related specialties that might affect their medical education. Medical students who learn in this environment might think choosing these specialties will mean better development and resources.

We also noticed that a higher ratio of P-IMGs underwent their specialty training in regional and local hospitals in nonurban areas. One possible explanation is that TMGs tend to receive specialist training in hospitals affiliated with medical schools. These hospitals are usually medical centers and are located in urban areas, and TMGs have generally spent 1 or 2 years in this hospital for clinical clerkship or internship. Furthermore, the professors at the medical school are often also the doctors in this hospital. The environment is quite familiar to them, and it is often an attractive career choice.

Conversely, P-IMGs have their medical education and clinical clerkship in other countries. Not long after they come back, they have to choose a specialist training location, and at this time, they perhaps do not know much about the medical system in Taiwan. They might have less affection for and familiarity with specific hospitals than TMGs.

The medical center has abundant resources, excellent teachers, treating a wide range of diseases and patients. Simultaneously, it also has relatively heavy work, estranged interpersonal relationships, and may be relatively unfamiliar with the simple diseases that primary doctors will face. Part of the P-IMGs willingness to serve in rural areas as primary doctors may be because regional hospitals can provide more suitable training. We have found similar circumstances in other countries. In the US, Canada, and New Zealand, IMGs are often more inclined to fill the gaps in workforce demands in underserved and rural areas than TMGs, or to serve socioeconomically disadvantaged populations [47,48,53,54,55,56,57]. Similarly, as mentioned earlier, the different medical and educational backgrounds between countries may cause a slight difference in their medical careers.

Combining the above two conclusions: no P-IMG has chosen those specialties that are highly competitive, and P-IMGs tend to work in regional and local hospitals in rural areas. There may be another explanation for this phenomenon. That is because certain highly competitive specialties such as dermatology, ophthalmology, etc. are often non-critical specialties, and may have few or even no training positions in smaller rural hospitals. If P-IMGs tend to apply for residency positions in small hospitals in rural areas, they might only have choices between those critical care specialties, such as internal medicine, surgery, etc. Specialties choices may be a compromise for the environment for them.

In addition, we have noticed that a certain percentage of P-IMGs tend to conceal their information from the public, especially about their graduate school [58]. One possible reason may be social distrust. Taiwanese society, especially after the fierce struggle in 2008–2009, has always distrusted and even despised P-IMGs. Most of them have strong family backgrounds [59] and sufficient financial support, and they are considered to have avoided the strict admission channels for Taiwan medical schools through loopholes. In addition, several adverse events have occurred in hospitals that are not favorable for them, and they are even considered to have poor attitudes and insufficient capabilities. Even today, the distrust of many medical circles and the public makes them conceal their actual academic qualifications. However, previous research has shown that carefully selected IMGs can also provide quality medical care [60,61,62] or even reduce mortality [63].

This research is the first analysis of P-IMGs to have taken place in Taiwan’s healthcare system for almost 20 years. The results showed some interesting and significant differences between P-IMGs and TMGs. This may enhance our understanding of this growing population and give a clearer view and information for those directors of hospitals and policymakers. On the other hand, it is never easy to study in a non-native language country, and it is difficult to face the adaptation and impact of language, culture, and change of living habits [14,64,65]. Being Asian may attract discrimination during certain times, such as the COVID-19 pandemic [66]. Thinking of the current situation of Taiwan’s medical system, where some specialties and areas are increasingly lacking a workforce [67], we should be more welcome to these P-IMGs and make the path returning home to practice medicine less difficult. Under a careful selection process, we believed they are also an excellent medicine force.

Although our study is the first research on P-IMGs, there are some limitations. First, because P-IMGs tend to hide their information, the data collected from the journal would not be intact. Therefore, there may be bias while analyzing these data. Second, we did not have data of pure TMGs for comparison, so most of the comparisons were made between P-IMGs and general medical graduates or P-IMGs and the overall residency training positions. Third, some P-IMGs may stay in Poland directly or go to practice medicine in other EU countries after graduation with appropriate language skills. Since we have no data, it remains unclear what career choices these IMGs make in different settings.

Much remains unknown. The comparison between P-IMGs and TMGs in the outcome of patients, credits they get, scholarly achievement, and even questionnaires for P-IMGs background, self-esteem, dilemma, and growth should be studied in the future. Conversely, following Philippines-trained IMGs, P-IMGs are the second-largest IMG population in Taiwan, followed by Myanmar-trained IMGs [10]. The differences in character and career choices between IMGs studying in different countries also deserve more understanding.

## 5. Conclusions

The number of P-IMGs has been increasing yearly, but they have suffered unequal treatment and even discrimination since the controversy in 2008–2009. They tend to choose less attractive specialties and receive specialist training in regional hospitals and urban areas. When Taiwan’s direct patient care-related specialties are relatively short of a professional workforce, recruiting physicians based on their talent rather than their background, through a careful selection process, is the direction that the Taiwan healthcare system should be working toward.

## Figures and Tables

**Figure 1 ijerph-19-03727-f001:**
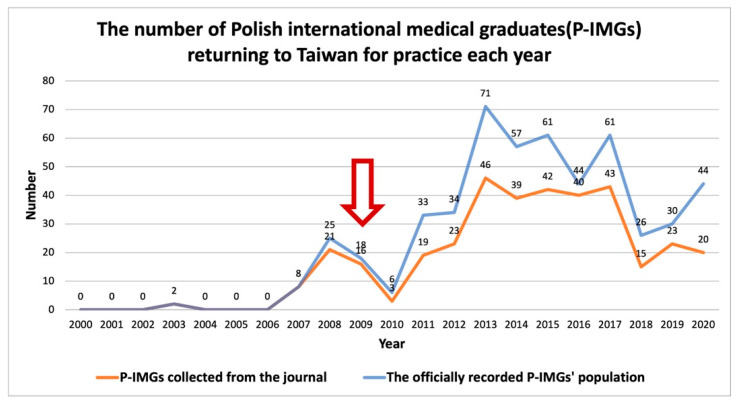
The number of Taiwanese who graduated from a Polish medical school (P-IMGs) and returned to Taiwan to practice each year. The red arrow represents the year that the mass demonstration between P-IMGs and domestic medical graduates was held.

**Figure 2 ijerph-19-03727-f002:**
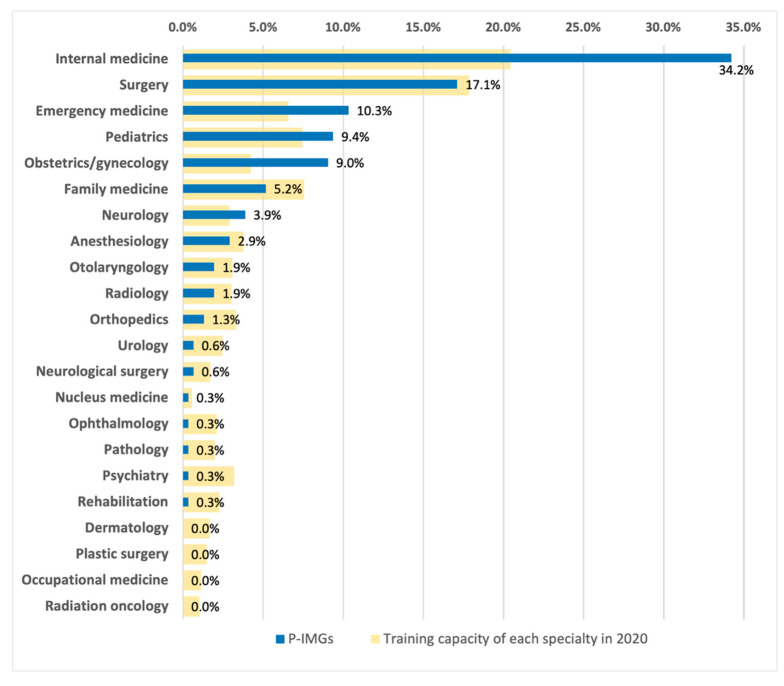
The distribution of the specialties chosen for training by Taiwanese who graduated from Polish medical schools (P-IMGs) between 2000–2020, compared with the training capacity of each specialty in a year (2020) in Taiwan.

**Table 1 ijerph-19-03727-t001:** The distribution of hospital levels chosen by Taiwanese students who graduated from Polish medical schools (P-IMGs) during 2000–2020, for their residency training, as well as the urbanization level of the area in which they were trained. (%, *n* = 306).

Urbanization Level	Residency in Medical Centers	Residency in Regional Hospitals	Residency in Local Hospitals	Total
1	75 (24.5)	31 (10.1)	2 (0.7)	108 (35.3)
2	76 (24.8)	55 (18)	3 (1)	134 (43.8)
3	0 (0)	22 (7.2)	0 (0)	22 (7.2)
4	0 (0)	42 (13.8)	0 (0)	42 (13.7)
Total	151 (49.3)	150 (49)	5 (1.7)	306 (100)

**Table 2 ijerph-19-03727-t002:** The distribution of all residency training positions in Taiwan in 2020 classified by the level of hospitals and the urbanization level of the area in which they are located. (%, *n* = 1610).

Urbanization Level	Specialty Training Ccapacity in Medical Centers	Specialty Training Capacity in Reginal Hospitals	Specialty Training Capacity in Local Hospitals	Total
1	659 (40.8)	122 (7.6)	4 (0.2)	785 (48.8)
2	564 (34.9)	166 (10.3)	1 (0.1)	731 (45.4)
3	0 (0)	31 (1.9)	1 (0.1)	32 (2)
4	0 (0)	62 (3.8)	0 (0)	62 (3.8)
Total	1223 (76)	381 (23.7)	6 (0.4)	1610 (100)

## Data Availability

Publicly available datasets were analyzed in this study. This data can be found here: [https://www.tma.tw/magazine/index.asp (accessed on 1 February 2022); https://www.tma.tw/stats/index_AllPDF.asp (accessed on 1 February 2022); https://dep.mohw.gov.tw/DOMA/fp-2713-46862-106.html (accessed on 1 February 2022)].

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
