# Peer review of "Choices of Specialties and Training Sites among Taiwanese Physicians Graduating from Polish Medical Schools"

_ijerph, 2022, doi:10.3390/ijerph19063727_

Round 1
Reviewer 1 Report
Thank you for giving me the possibility of riding this interesting manuscript. I have some comments to make it even better.
Introduction
Line 48 “This fact and the belief that doctors are well paid and have high social status still prevail in our society.” Please support this statement with reference.
Lines 74-75 “Compared with those in the US and Europe, their medical schools targeting international students have low entrance requirements and much lower tuition fees, coupled with a lower cost of living.” Please support this statement with references.
You should add information about the Polish medical education system – how many years? how many medical schools? how many international students? Please see the recent publications from IJERPH where you can find basic information:
JERPH | Free Full-Text | A Qualitative Study of the Mistreatment of Medical Students by Their Lecturers in Polish Medical Schools (mdpi.com)
IJERPH | Free Full-Text | A Study of Differences in Compulsory Courses Offering Medicine Humanization and Medical Communication in Polish Medical Schools: Content Analysis of Secondary Data (mdpi.com)
Materials and Methods
Is the database authors created is available publicly? If yes, please state where.
Lines 187-188 – please add at the end something like… “so the Ethics Committee approval was not obtained”.
Results
Descriptive statistics - that's a bit too little for a good article. Please consider using e.g. Ch2 test, to see if there is the statistical dependency between the data; “P-IMGs chose to be trained in were internal medicine (34.2%), surgery (17.1%), emergency medicine (10.3%), pediatrics (9.4%), and obstetrics/gynecology (9.0%), which accounted for 80% of P-IMGs in total; while in a year in Taiwan, the training positions of these five specialties account for only 56.5% of total residency training positions, 20.4%, 17.9%, 6.6%, 7.4% and 4.2%, respectively.”
Discussion
I think that it is also worth emphasizing in the discussion that Asian students could also encounter difficulties in Poland while studying medicine. Generally speaking, it is not easy for them, neither in Poland nor after their return home. Please see:
COVID-19-related prejudice toward Asian medical students: A consequence of SARS-CoV-2 fears in Poland - ScienceDirect
Reviewer 2 Report
Dear Authors,
thank you for submitting the manuscript and for the opportunity to review it.
The article "Choices of specialties and training sites among Taiwanese physicians graduating from Polish medical schools" deals with a topic relevant to health policy and is written in an understandable way. The data themselves are evaluated and presented in a well-structured manner. The literature cited is up to date and relevant to the context (although often limited in readability due to the Chinese language).
Biggest limitation for me would be that the article itself describes a very local problem (which is also evident in the linked literature). So, in my eyes, the biggest challenge is to get the international readership interested in the topic.
Therefore, I have some specific but also general comments/suggestions on this:
- The introduction is well and understandably written and sufficient in scope. The first section briefly discusses the significance of IMG in other countries. The last and largest section is devoted to the debate on P-IMGs. Perhaps here (or at the latest in the discussion) a look at other countries where there have been similar controversies would be conceivable.
- The Materials and Methods section describes the selection of data. The question here would be whether all physicians working in Taiwan are basically members of the Taiwan Medical Association or whether there are exceptions here that are not covered? Or is there a larger proportion of physicians who come to Taiwan from other countries to study, or physicians who were born and studied in other countries and then work in Taiwan?
- In section 2.4, the information about the manufacturer and the exact program version should usually be added for Microsoft Excel.
- The data analysis was done from 2000-2020 - consequently, I would also start in 2000 on the X-axis in Figure 1.
- In Table 1a and 1b, I would write the total number (n=306 and n=1610, respectively) in each table label and delete it from the columns. The information in the header of each column is rather confusing. Maybe you can support the type of crosstab by adjusting the formatting in which the row/column "Total" is separated by a thicker black line or a different background color.
- Another point would be important for the interpretation and evaluation of the data. Is there any data on how many high school graduates from Taiwan go on to study medicine abroad overall, or even specifically in Poland? The publication breaks down the doctors who came back to Taiwan after graduation. But there will also be a larger proportion of graduates who stayed in the country where they studied?
Or have changed the country again. Particularly in the European Union, medical degrees are largely recognized between countries, so that graduates from Poland can also move to Spain, France, Italy or Germany to work (after taking appropriate language courses). - The discussion also talks about the specialties that are particularly popular in Taiwan (dermatology, ophthalmology, etc.). The question here would be how "popular" is defined. It is certainly not the subjects that are chosen by most graduates for further training, but the specialties where the ratio of further training places and demand is most unfavorable, isn't it?
- It can be assumed that certain specialties such as dermatology, ophthalmology, etc. are also only offered in the large hospitals in the big cities and not in small hospitals in the countryside. If P-IMGs are now working more in the smaller rural hospitals it does not seem surprising that there is less demand for these specialties. Here is the cause/effect question. In addition, the question would be whether P-IMGs are then not selecting such specialties in Poland for further training.
- Finally, the health policy question for the discussion would be whether the increased use in rural regions could not even be used sensibly. In Europe, too (e.g. in Germany), there is a shortage of doctors to care for the population, especially in rural regions, and in some cases there is an oversupply in the large cities. In order to counteract this, there are concepts that support the medical profession in rural regions. For example, through financial support during studies or through a special quota in the allocation of study places. The students must then commit themselves to working in rural areas for a few years after completing their studies. Similar concepts exist in other countries. Perhaps this could be looked at again and seen more as an opportunity.
I hope that I could contribute positively to a revision of the manuscript with my comments and wish you much success!
Kind regards!
Round 2
Reviewer 1 Report
Accept in present form
Reviewer 2 Report
Dear authors,
Thank you very much for the explanations and the revision of the manuscript, which in my opinion has been significantly improved. I have no further comments.
Kind regards!